# Phacoemulsification combined with goniosynechialysis versus phacoemulsification alone for patients with primary angle-closure disease: A meta-analysis

**Lin Yao[1‡], Haitao Wang[2‡], Yunxiao Wang[2‡], Pengpeng Zhao[1], Haiqing Bai [2]***

**1** Qingdao Aier Eye Hospital, Qingdao, China, **2** Department of Ophthalmology, The Affiliated Hospital of Qingdao University, Qingdao, China

‡ LY, HW and YW contributed equally to this work as first authors.
* haiqing_bai@126.com, haiqingbai@qdu.edu.cn

## Abstract

This meta-analysis aims to systematically compare the efficacy between phacoemulsification (PE) combined with goniosynechialysis (GSL) and PE alone for primary angle-closure disease (PACD) patients. All the data were searched from the PubMed, EMBASE and the Cochrane Library. The Cochrane Handbook was used to evaluate the quality of the included studies. Additionally, this meta-analysis was performed by using the Revman 5.4 software. Nine randomized controlled trials (RCTs) were included in this study. Compared with PE alone group, PE+GSL could result significant reduction in the IOP (MD, 1.81; p = 0.002). In the instrumental subgroup, also more reduction of IOP was shown in the PE+GSL group (MD, 2.11; p = 0.02). In the viscogonioplasty (VGP) subgroup, there was not no statistical difference between PE alone group and PE+GSL group (MD, 1.53; p = 0.11). Also, more reduction of peripheral anterior synechiae (PAS) was shown in the PE+GSL group (MD, 59.15; p<0.00001). For the change in angle open distance (AOD)500, AOD 750, trabecular-iris space (TISA)500, number of glaucoma medications and best corrected visual acuity (BCVA), there was no difference between two groups (p = 0.25, 0.35, 0.17, 0.56, 0.08). For TISA 750, more improvement was shown in the PE+GSL group (p<0.00001). Instrumental separation had better effect on lowering IOP when it combined with PE. Both instrumental separation and VGP could reduce postoperative PAS. The operation of GSL has no obvious effect on postoperative vision.

## Introduction

Primary angle-closure glaucoma (PACG) usually can cause irreversible damage to vision. PACG is worldly estimated to affect 15 million people. It is more common amongst Asians, especially Chinese [1–3]. The blockage of the aqueous drainage due to the peripheral anterior

**Data Availability Statement:** All relevant data are within the paper.

**Funding:** The authors received no specific funding for this work.

**Competing interests:** The authors have declared that no competing interests exist.

synechiae (PAS), which finally results in the raising intraocular pressure (IOP) [4]. Crowed anterior chamber and iridotrabecular meshwork contact are the predisposition for PAS [4].

In eyes with angle closure, lens extraction by phacoemulsificatoin (PE) can deepen the anterior chamber and reverse the mechanism of the angle-crowding, meanwhile, decrease the iridotrabecular contact, widen the angle and improve aqueous outflow facility. Therefore, PE can lower the IOP for patients with PACG [5, 6]. Furthermore, some studies have shown that the lowering IOP depended on the reduction of PAS [7, 8]. Goniosynechialysis (GSL) which can be performed combined with PE, is a surgical technique to break the PAS [9–14]. It involves the physical separation of PAS by using surgical instrument under direct visualization with goniolens or by cohesive viscoelastic without direct visualization. The latter technique is also called viscogonioplasty (VGP) [6, 15, 16].

Recent years, many studies have demonstrated that both PE combined with GSL (PE+GSL) and PE alone can effectively lower the IOP for patients with primary angle-closure (PAC) or PACG. Correspondingly, some of these studies compared these two surgeries [6, 9–16]. The primary objective of this meta-analysis is to compare the efficacy between PE+GSL and PE alone for patients with PAC or PACG.

## Materials and methods

This meta-analysis of prospective randomized clinical trials (RCTs) was performed according to the Preferred Items for Systematic Reviews and Meta-Analyses (PRISMA) guidelines.

### Methods of literature search

Computer databases including PubMed, EMBASE and Cochrane Library were searched from the year 2000 to January 2023. Year 2000 was selected to be consistent with the way PE surgeries were performed. Medical Subject Headings (MeSH) and free words closely related with PE and GSL were used. (((phaco*[Title/Abstract]) OR (phacoemulsification[Title/Abstract])) OR (phako*[Title/Abstract])) AND ((((viscogonioplasty[Title/Abstract]) OR (goniosynechialysis [Title/Abstract])) OR (GSL[Title/Abstract])) OR (VGP[Title/Abstract])) was used for searching from the PubMed.

**Inclusion criteria.**  1.  RCTs.

2. Patient populations with PAC or PACG.

3. All eyes were treated with PE with or without GSL.

4. At least one primary or secondary outcome reported.

5. Follow-up time was more than one month.

6. Studies published in English.

**Exclusion criteria.**  1.  Non-randomized studies.

2. Cohort studies, case report, reviews and editorial.

3. Studies without outcomes of interest.

### Data extraction

Two reviewers (LY and HB) independently extracted data using the pre-established extraction tables, including the following: (1) Basic characteristics of the study, including the name of the first author, year of publication, and follow-up time (2) Basic characteristics of the patients,

including the mean age and standard deviation (SD) (3) Baseline and post-operative characteristics such as IOP, range of PAS, AOD, TISA, number of glaucoma medications and BCVA. If consensus was not achieved by these two reviewers, the third reviewer would be intervened to provide a decision [17–20].

## Outcome measures

The primary outcomes were mean change in IOP and PAS. The secondary outcomes were angle open distance (AOD) at 500, 750μm and trabecular-iris space (TISA) at 500, 750μm evaluated by anterior segment OCT, mean change in number of glaucoma medications from baseline and best corrected visual acuity (BCVA).

## Qualitative assessment

The quality of the included studies was assessed by the Cochrane Handbook, including 6 items: random sequence generation, allocation concealment, blinding of participants and personnel, blinding of outcome assessment, incomplete outcome data, selective reporting, and other biases. Two reviewers determined the risk of bias which had three options (low, high, and unclear) [21]. If consensus was not achieved by these two reviewers, the third reviewer would be intervened to provide a decision. When necessary, we contacted the authors of the studies to obtain the full text or related information for an accurate assessment [17–20].

## Statistical analysis

Review Manager (version 5.4; Cochrane Collaboration) was used for data analysis. The statistical heterogeneity of the included studies was test by the Cochrane $I^2$ test. A random-effect model was applied. Mean difference (MD) with a 95% confidence interval (CI) was used to estimate the effectiveness for continuous variable data [17–20].

Subgroup analyses were performed according to whether the GSL performed by using surgical instrument (instrumental subgroup) or viscoelastic (VGP subgroup).

## Results

A total of 96 studies were retrieved. Finally, 9 RCTs were included in this meta-analysis. The basic characteristics of the included studies were shown in Table 1. There were 237 patients in the PE+GSL group and 229 patients in the PE alone group. In six studies, surgical instrument was used to physically break the PAS under gonioscopic visualization in the PE+GSL group. In other three studies, patients in the PE+GSL group were received VGP, a cohesive viscoelastic was injected near the angle following intraocular lens (IOL) implantation to break the PAS. The literature screening process was shown in Fig 1.

### Methodological quality evaluation

The results of the methodological evaluation according to the Cochrane Handbook were shown in Fig 2.

### Efficacy analysis

**Mean change in IOP from baseline.** All nine studies reported changes in IOP from baseline. More reduction of IOP was shown in the PE+GSL group (MD, 1.81; p = 0.002). In the instrumental subgroup, also more reduction of IOP was shown in the PE+GSL group (MD, 2.11; p = 0.02). However, in the VGP subgroup, there was not statistical difference between the PE+GSL group and PE alone group (MD, 1.53; p = 0.11) (Fig 3).

**Table 1. Basic characteristics of include studies.**

| Auther (Year) | Study Location | Intervention | Technique (Subgroup) | N | Follow-up (months) | Mean Age | Age (SD) | Male/ Female | Baseline IOP (Mean) | Baseline IOP (SD) | Post-Operative IOP (mean) | Post-Operative IOP (SD) |
|---|---|---|---|---|---|---|---|---|---|---|---|---|
| Angmo D 2019[9] | India | PE+GSL | Instrument | 34 | 6 | 56.50 | 9.17 | None | 30.72 | 3.88 | 13.21 | 1.97 |
| | | PE | | 30 | | 58.77 | 8.14 | | 29.48 | 6.76 | 13.17 | 1.66 |
| Husain R 2019[10] | Singapore | PE+GSL | Instrument | 38 | 12 | 68.1 | 9.2 | 13/25 | 22.9 | 5.3 | 15.9 | 4.5 |
| | | PE | | 40 | | 67.3 | 8.6 | 11/29 | 22.3 | 8.5 | 14.3 | 5.0 |
| Lee CK 2015[11] | Korea | PE+GSL | Instrument | 15 | 2 | 66 | - | 11/4 | 15.87 | 4.02 | 13.53 | 2.80 |
| | | PE | | 15 | | 64 | - | 14/1 | 11.33 | 2.50 | 11.20 | 2.54 |
| Rodrigues IA 2017[12] | UK | PE+GSL | Instrument | 13 | 6 | 67.2 | 8.4 | 5/8 | 27.4 | 7.2 | 14.8 | 2.7 |
| | | PE | | 10 | | 66.1 | 7.4 | 5/5 | 19.6 | 5.5 | 14.2 | 3.1 |
| Shao T 2015[13] | China | PE+GSL | Instrument | 23 | 6 | 73.61 | 8.44 | None | 22.12 | 5.98 | 13.65 | 2.46 |
| | | PE | | 20 | | 69.85 | 8.56 | | 23.45 | 7.99 | 16.40 | 5.58 |
| Tun TA 2015[14] | Singapore | PE+GSL | Instrument | 11 | 12 | 66.75 | 6.53 | 3/8 | 21.82 | 5.81 | 14.36 | 2.98 |
| | | PE | | 11 | | 67.77 | 5.18 | 2/9 | 18.73 | 6.54 | 16.91 | 5.72 |
| Eslami Y 2013[15] | USA | PE+GSL | VGP | 33 | 1.5 | 64.3 | 9.8 | 11/16 | 24.5 | 6.8 | 16.9 | 4.9 |
| | | PE | | 32 | | 65.4 | 8.4 | 13/15 | 21.6 | 6.0 | 15.6 | 5.6 |
| Moghimi S 2015[16] | Iran | PE+GSL | VGP | 45 | 12 | 61.6 | 8.3 | 19/26 | 23.3 | 7.3 | 14.5 | 2.5 |
| | | PE | | 46 | | 63.2 | 6.9 | 19/27 | 22.3 | 6.3 | 14.0 | 3.7 |
| Varma D 2010[6] | UK | PE+GSL | VGP | 25 | 12 | 72.40 | 8.9 | 16/9 | 30.12 | 7.03 | 13.7 | 2.89 |
| | | PE | | 25 | | 72.96 | 7.8 | 8/17 | 29.68 | 8.73 | 16.2 | 3.55 |

GSL, goniosynechialysis; VGP, viscogonioplasty; PE, phacoemulsification; IOP, intraocular pressure;

**Mean change in PAS.** Six studies reported changes in PAS. The overall MD was 59.15 (95%CI, 36.28 to 82.03), compared with PE alone group more reduction of PAS was shown in the PE+GSL group (p<0.00001). As well, in the instrumental subgroup and VGP subgroup, more reduction of PAS was shown in the PE+GSL group (MD, 53.32, 66.00; 95%CI, 5.96 to 100.67, 43.13 to 88.88; p = 0.03, <0.00001) (Fig 4).

**Mean change in number of glaucoma medications.** Eight studies reported the change in number of glaucoma medications before and after surgery. The overall MD was 0.08 (95%CI, -0.19 to 0.35), and there was no statistical difference between the PE+GSL group and PE alone group (p = 0.56). As well, in the instrumental subgroup and VGP subgroup analysis, there was not statistical difference between the PE+GSL group and PE alone group (MD, 0.05, 0.20; 95% CI, -0.32 to 0.42, -0.12 to 0.52; p = 0.79, 0.22) (Fig 5).

**Mean change in AOD 500 and AOD 750.** Three studies reported the change in AOD before and after surgery. The overall MD was 0.16 and 0.21 (95%CI, -0.12 to 0.45, -0.23 to 0.66), and there was no statistical difference between the PE+GSL group and PE alone group in both AOD 500 and 750 (p = 0.25 and 0.35). As well, in the instrumental subgroup and VGP subgroup, there was not statistical difference between the PE+GSL group and PE alone group

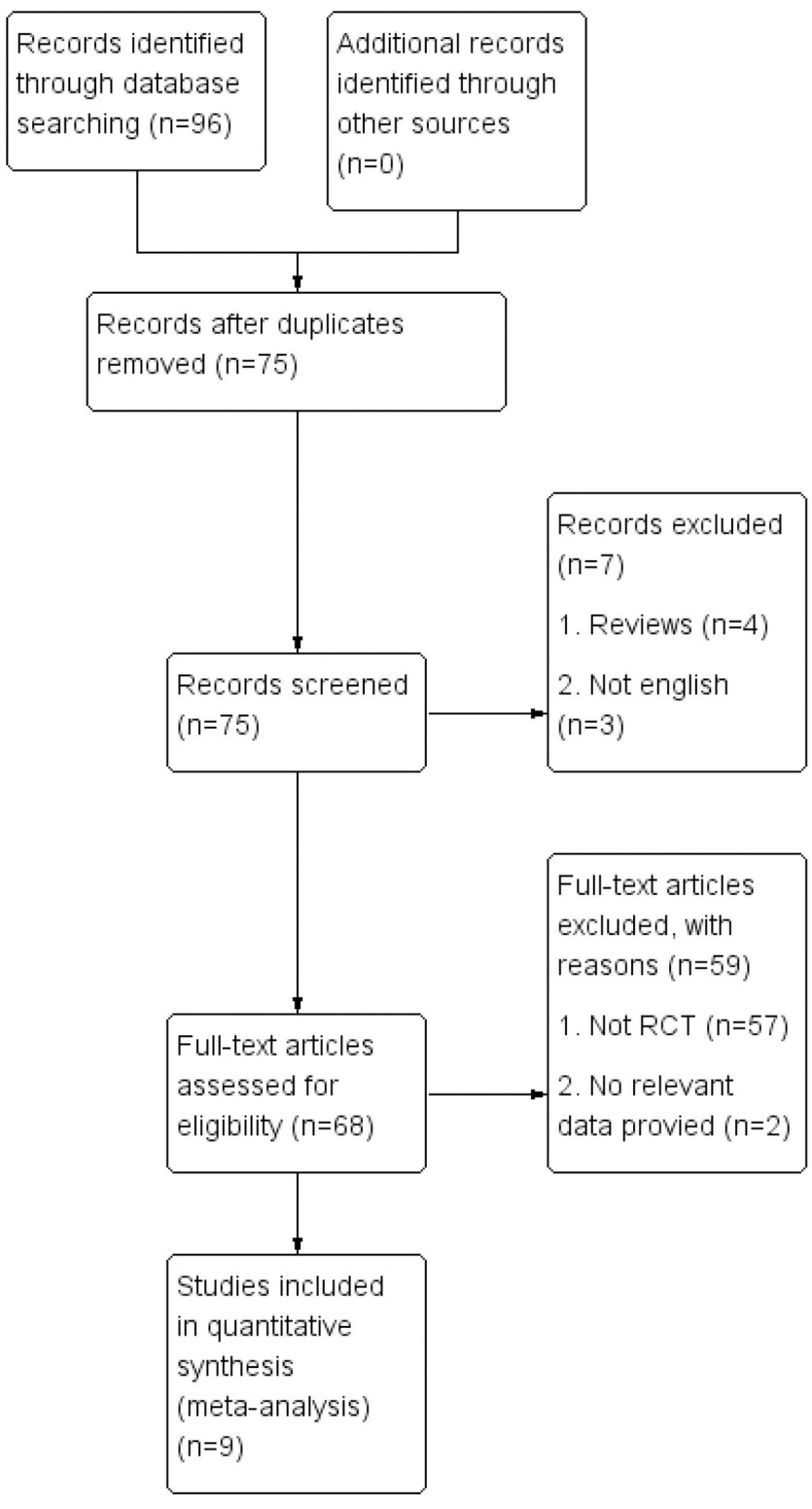

**Fig 1. Prisma flow diagram.**

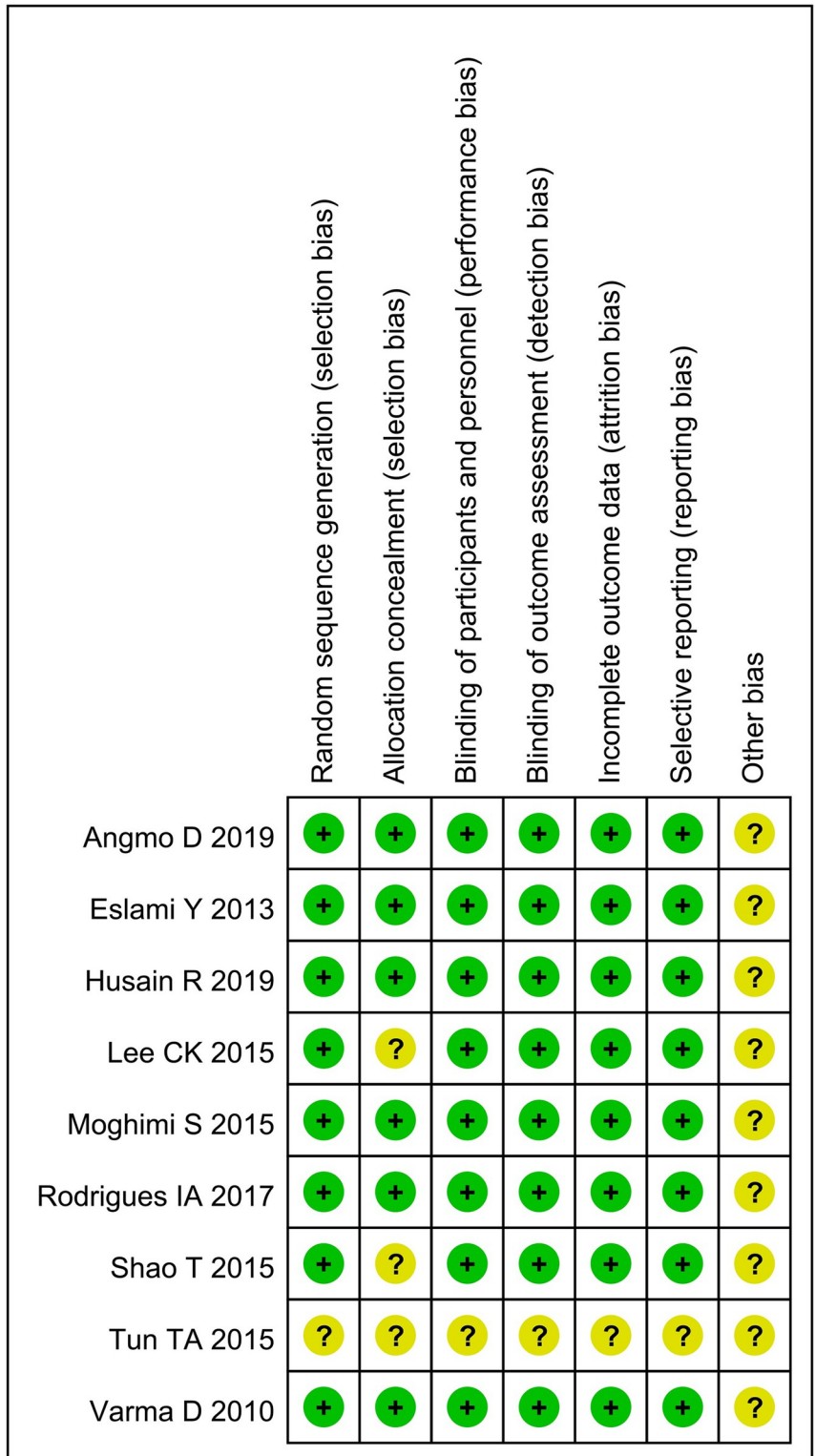

**Fig 2. The results of the methodological evaluation.**

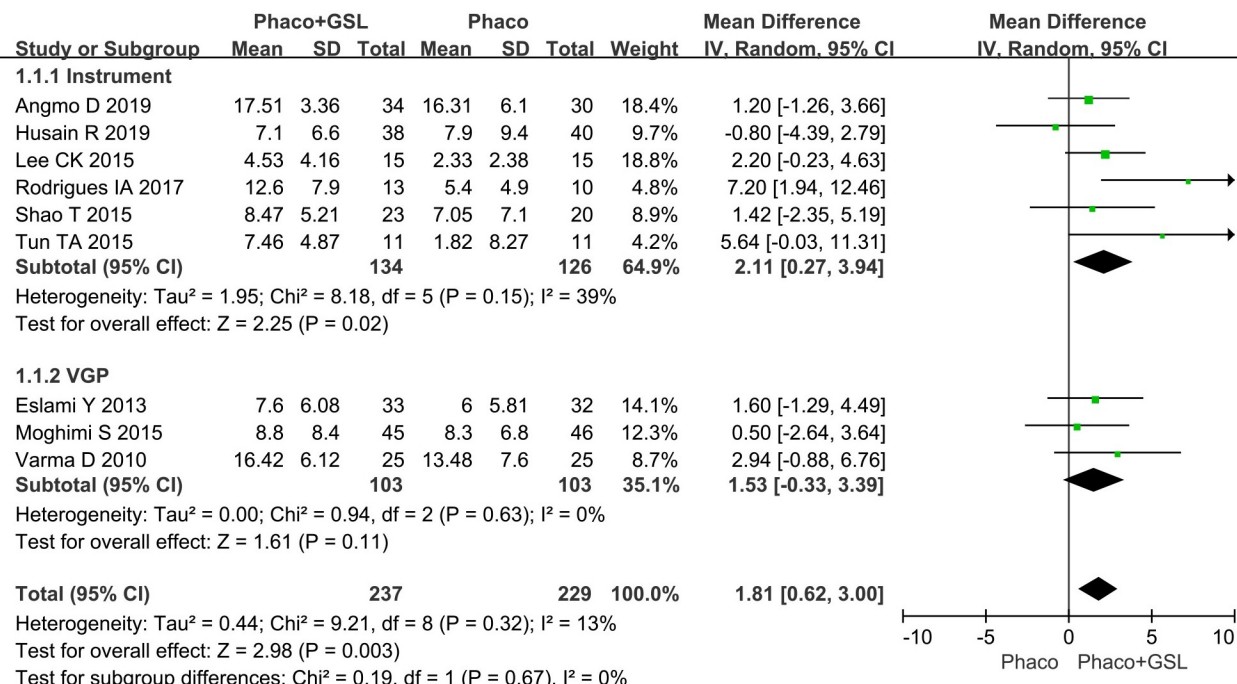

**Fig 3. Forest plot for mean change in IOP from baseline.** SD indicates standard deviation, CI indicates confidence interval.

(AOD 500, MD, 0.24, 0.03; 95%CI, -0.08 to 0.55, -0.02 to 0.09; AOD 750, MD, 0.30, 0.03; 95% CI, -0.29 to 0.89, -0.04 to 0.10; p = 0.31,0.44) (Figs 6 and 7).

**Mean change in TISA 500 and TISA 750.** Four studies reported the change in TISA 500 and five studies reported the change in TISA 750. For TISA 500, there was no statistical

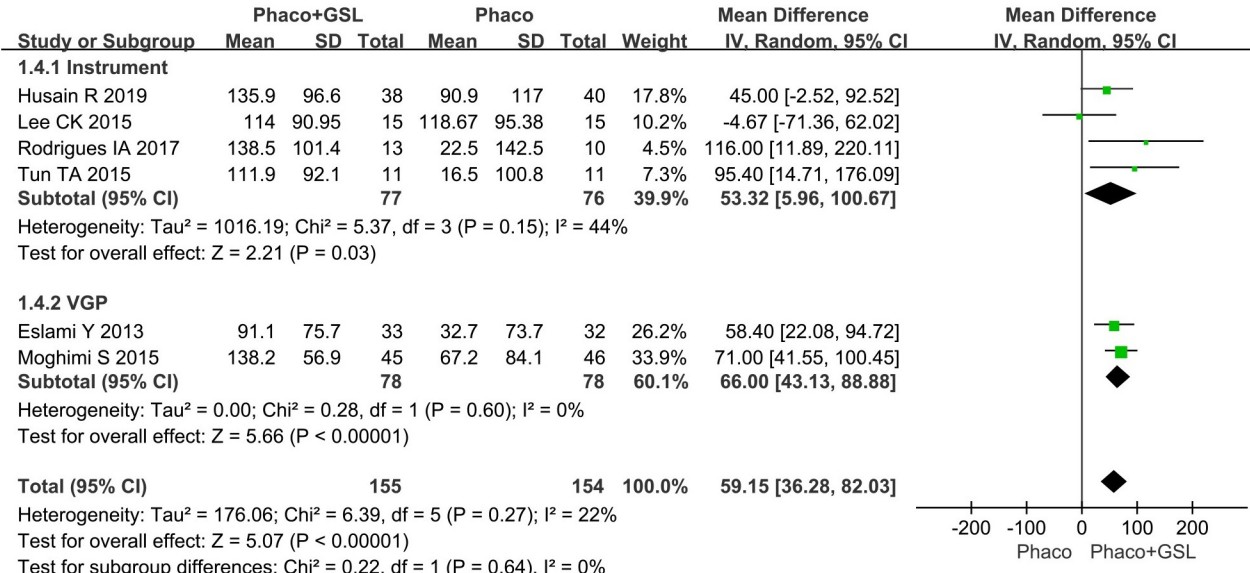

**Fig 4. Forest plot for mean change in PAS.** SD indicates standard deviation, CI indicates confidence interval.

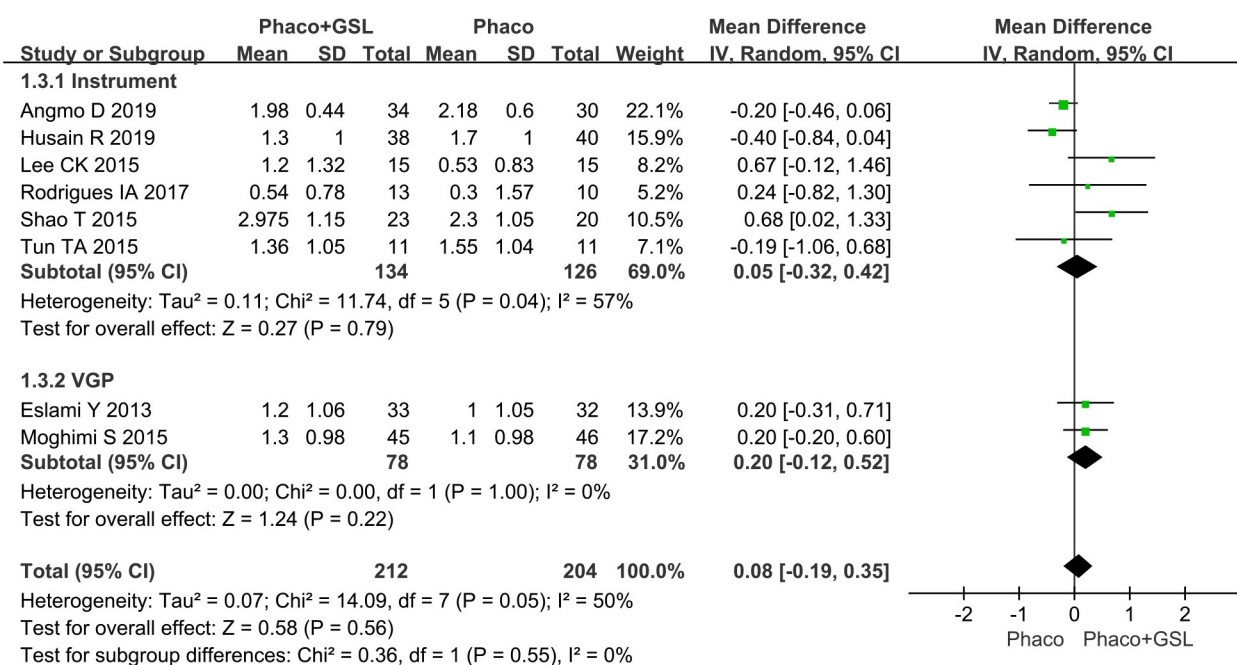

**Fig 5. Forest plot for mean change in number of glaucoma medications.** SD indicates standard deviation, CI indicates confidence interval.

difference between the PE+GSL group and PE alone group (MD, p = 0.17). As well, in the instrumental subgroup and VGP subgroup, there was no statistical difference between the PE +GSL group and PE alone group (MD, 0.07, 0.02; 95%CI, -0.03 to 0.18, -0.01 to 0.04; p = 0.16,0.13) (Fig 8). For TISA750, more improvement was shown in the PE+GSL group (MD, 0.10; p<0.00001). As well, in the instrumental subgroup and VGP subgroup, more improvement was shown in the PE+GSL group (MD, 0.17, 0.03; 95%CI, 0.15 to 0.19, 0.01 to 0.05; p<0.00001, p = 0.008) (Fig 9).

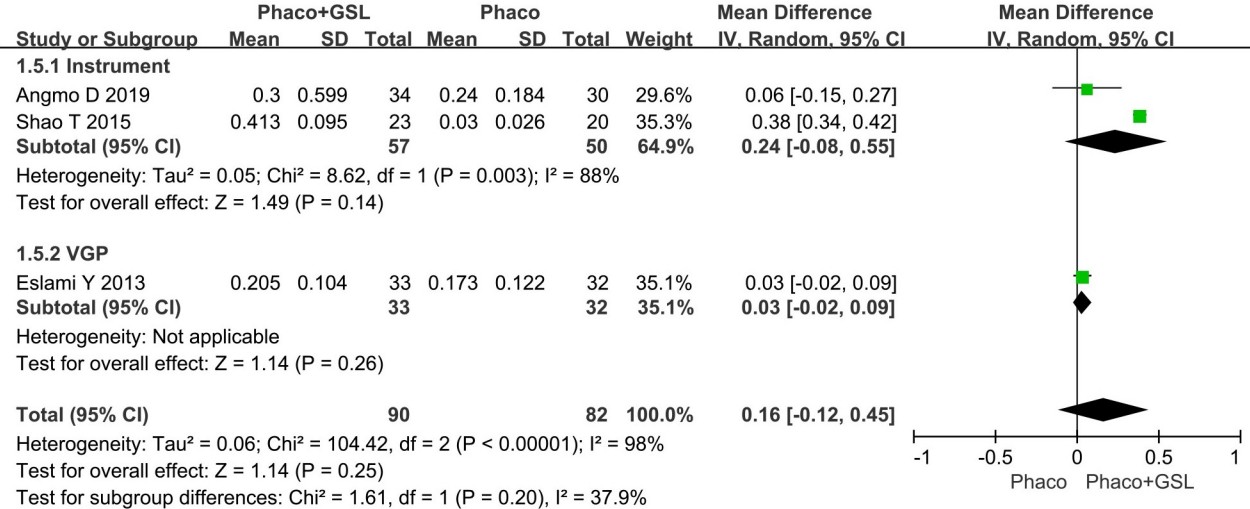

**Fig 6. Forest plot for mean change in AOD 500.** SD indicates standard deviation, CI indicates confidence interval.

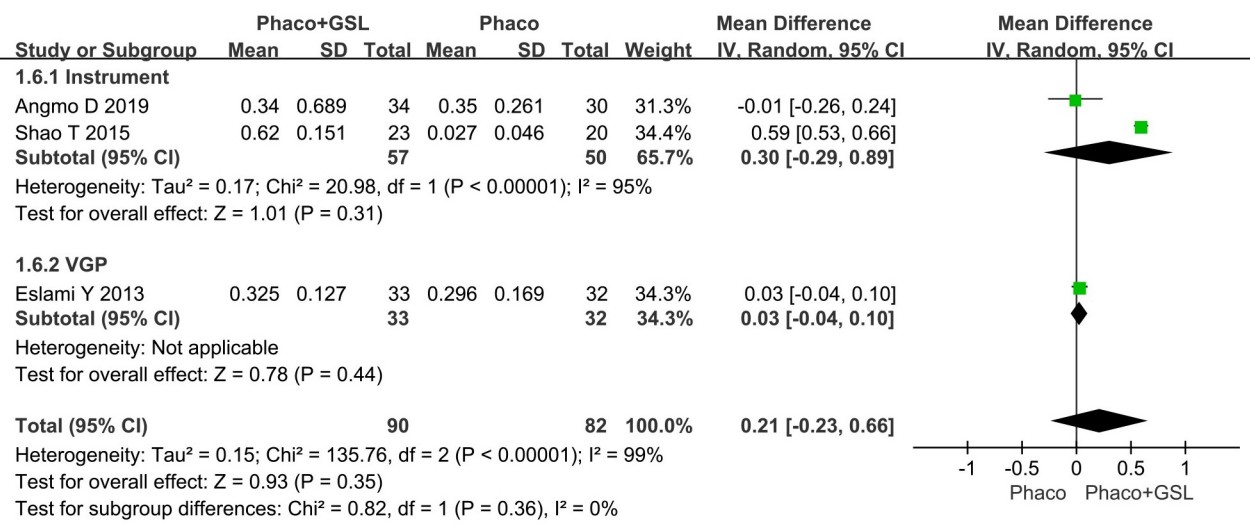

**Fig 7. Forest plot for mean change in AOD 750.** SD indicates standard deviation, CI indicates confidence interval.

**Mean change in BCVA from baseline.** Six studies reported changes in BCVA from baseline. The overall MD was 0.03 (95%CI, -0.03 to 0.09), and there was no statistical difference between the PE+GSL group and PE alone group (p = 0.29). As well, in the instrumental subgroup and VGP subgroup, there was not statistical difference between two groups (MD, 0.05, 0; 95%CI, -0.03 to 0.13, -0.09 to 0.09; p = 0.21,0.96) (Fig 10).

## Discussion

According to the International Society of Geographic and Epidemiologic Ophthalmology (ISGEO) classification, PAC had led to significant glaucomatous damage to the optic nerve would be defined as PACG [22]. Therefore, both PAC and PACG should be classified as

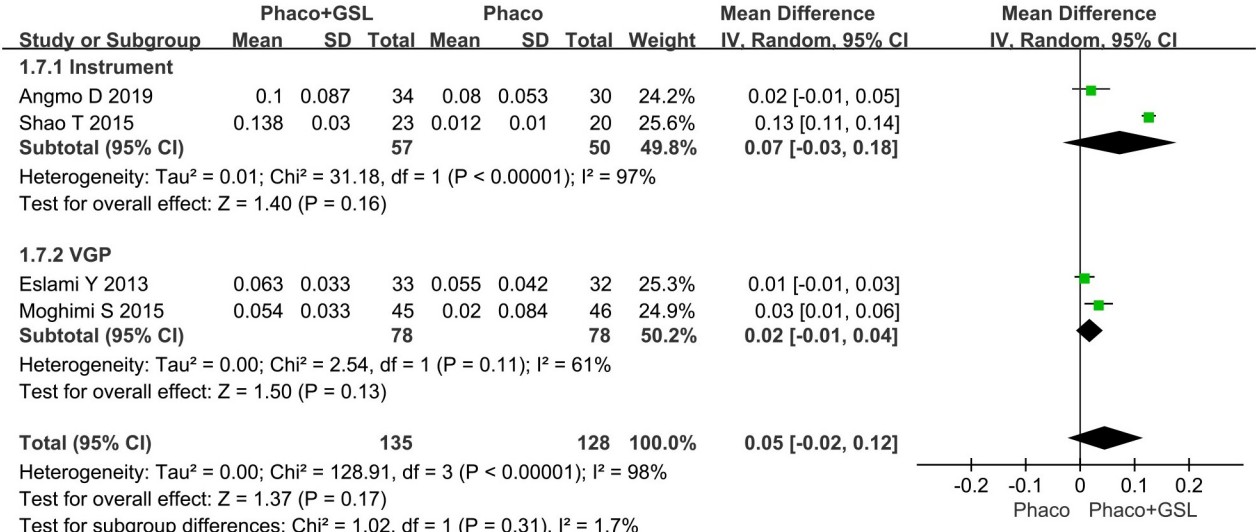

**Fig 8. Forest plot for mean change in TISA 500.** SD indicates standard deviation, CI indicates confidence interval.

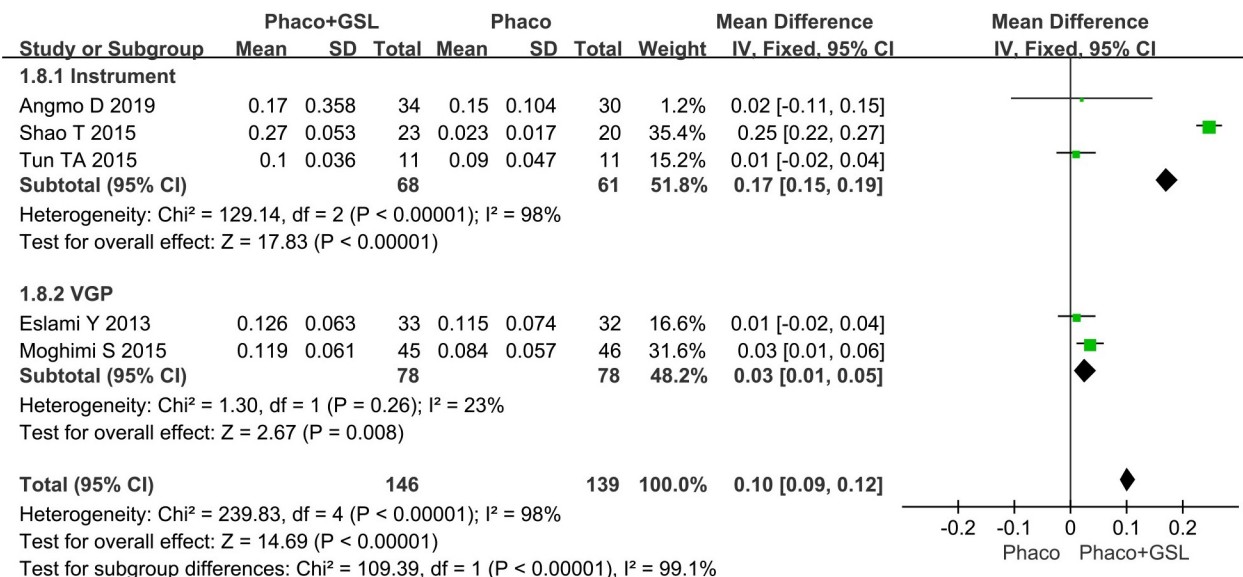

**Fig 9. Forest plot for forest plot for mean change in TISA 750.** SD indicates standard deviation, CI indicates confidence interval.

primary angle-closure disease (PACD). In our meta-analysis, seven from nine included RCTs focused on PACG [6, 9, 11, 13–16], and other two RCTs focused on PAC [10, 12].

Previous studies proved that both PE+GSL and PE alone could effectively reduce the IOP of patients with PACD [6, 9–16]. Therefore, the potential different effect on lowering IOP derived from PE+GSL and PE alone was the primary issue of concern. Most of the included studies concluded that there was no statistical difference in lowering IOP between the PE+GSL and PE alone group [6, 9–11, 13–16]. However, as all nine included studies were merged into

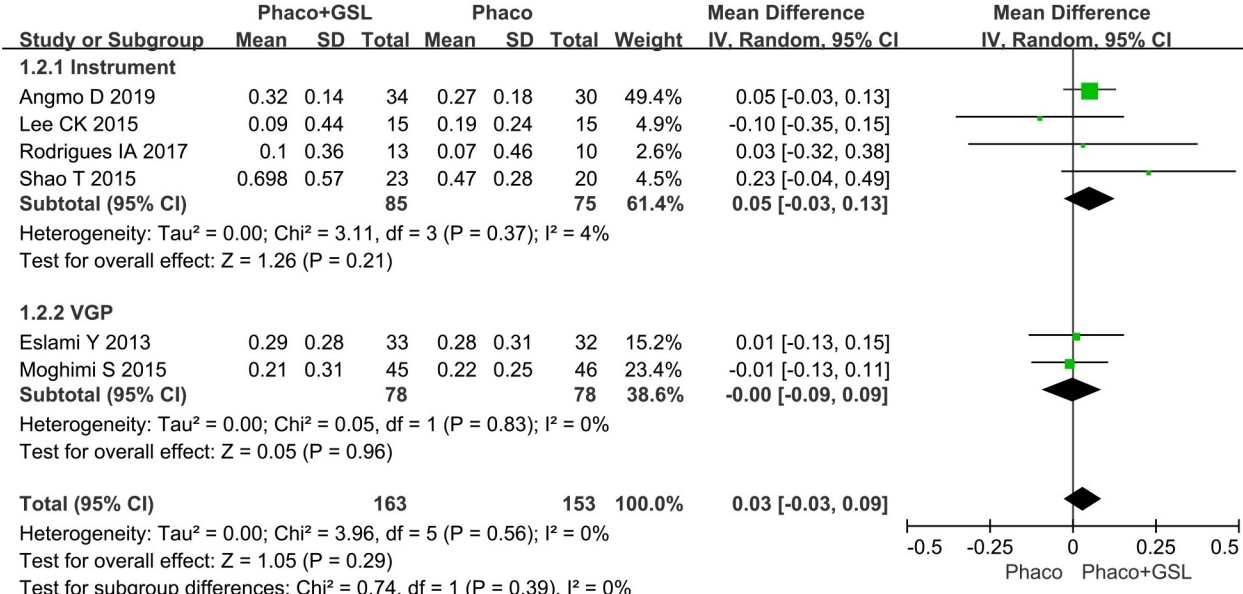

**Fig 10. Forest plot for mean change in BCVA from baseline.** SD indicates standard deviation, CI indicates confidence interval.

an overall data, the meta-analysis provided that PE+GSL had better effect of lowering IOP compared with PE alone. As well as in the instrumental subgroup, PE+GSL had better effect of lowing IOP. However, in the VGP subgroup, there was no significant difference in reduction of IOP between PE+GSL and PE alone. This was consistent with all the three included studies focused on VGP in our meta-analysis [6, 15, 16]. Therefore, the combination of VGP in PE may not bring extra effect of lowering IOP.

Some studies showed that the lowering IOP depended on the change of PAS extent [7, 8]. The trabecular meshwork could be exposed again by separating PAS. Furthermore, the separating PAS and possible restoration of trabecular function before irreversible structural changes, which seem to be a logical approach to reduce IOP of PACG patients [23, 24]. Previous studies showed that the decreasing of PAS was significant after performing both PE+GSL and PE alone [10–12, 14–16]. In this meta-analysis, we further analysed the effect of additional GSL in PE on the PAS extent. The change of PAS extent in the PE+GSL group was significantly greater than PE alone group. In the subgroup analysis, both the instrumental subgroup and VGP subgroup showed the similar results. Previous research revealed that mechanical deepening of the anterior chamber with viscoelastic or saline infusion during PE alone may also open some PAS [25]. Obviously, PE+GSL has better effect of less PAS. After all, GSL is specifically used to break PAS. We found both instrumental separation and VGP could decrease PAS. For the effect of lowering IOP, VGP combined with PE did not seem to bring additional effect compared with PE alone. The reasons may be as follows. The trabecular meshwork which behind the PAS may not be functional. Therefore, the mechanical separating PAS may not result in reduction of IOP. Moreover, gonioscopic appearance might hardly reflect the function of the trabecular meshwork. The loss of trabecular cells as well as the irregular architecture of the trabeculum which might exist in areas away from visible PAS or under the PAS [26]. However, instrumental separation could bring additional lowering IOP. This may be because the mechanical separation by using surgical instrument under direct visualization is clearer and stronger than VGP. In addition to separation of the PAS, the trabecular meshwork may also be pulled during the separation process, which may make the outflow of aqueous easier.

There was no significant difference in most parameters of anterior chamber angle (ACA) between the two groups before and after surgery. Only TISA 750 was different between the two groups. Therefore, GSL may not have much influence on ACA.

Eight involved studies revealed that there was no significant difference in reducing glaucoma medications between the PE+GSL group and PE group [9–16]. Our meta-analysis also confirmed that there was no significant difference in reducing glaucoma medications between two groups. Furthermore, subgroup analyses obtained the same result.

Six included studies concluded that there was no significant difference in improvement of BCVA between the PE+GSL group and the PE group [9, 11–13, 15, 16]. Our meta-analysis also confirmed that no significant difference could be found in improvement of BCVA between two groups. In further subgroup analyses, there was also no significant difference in improvement of BCVA. These results illustrated that the operation of GSL may have no effect on the prognosis of postoperative vision.

GSL is a safe surgical procedure. Compared with Phaco alone, the possible risk of Phaco +GSL is hyphema, which was reported in three included studies [9, 12, 13].

This meta-analysis also has certain limitations. In the majority, this meta-analysis was conducted in Asian populations. In all nine included studies, six studies were from Asia. It is largely known that can have different patterns of anatomic characteristics in this population when compared to Caucasian or Hispanic origin population.

## Conclusions

Both instrumental separation and VGP could reduce postoperative PAS. However, it seems that only instrumental breaking the PAS could bring extra effect of lowering IOP when it combined with PE. And PE combined with or without GSL has no influence on postoperative vision.

## Supporting information

**S1 Checklist. PRISMA 2020 checklist.**
(DOCX)

## Author Contributions

**Conceptualization:** Lin Yao, Haitao Wang.

**Data curation:** Lin Yao, Haitao Wang, Yunxiao Wang, Pengpeng Zhao.

**Formal analysis:** Lin Yao.

**Investigation:** Lin Yao, Yunxiao Wang.

**Methodology:** Lin Yao, Yunxiao Wang, Pengpeng Zhao.

**Supervision:** Lin Yao, Haiqing Bai.

**Writing – original draft:** Lin Yao.

**Writing – review & editing:** Lin Yao, Haiqing Bai.

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
