## [Decision Letter · Decision Letter 0]

26 Oct 2023

PONE-D-23-24472Phacoemulsification combined with goniosynechialysis versus phacoemulsification alone for patients with primary angle-closure disease: a meta-analysisPLOS ONE

Dear Dr. Bai,

Thank you for submitting your manuscript to PLOS ONE. After careful consideration, we feel that it has merit but does not fully meet PLOS ONE’s publication criteria as it currently stands. Therefore, we invite you to submit a revised version of the manuscript that addresses the points raised during the review process.

We look forward to receiving your revised manuscript.

Kind regards,

Andrzej Grzybowski

Academic Editor

PLOS ONE

Reviewers' comments:

Reviewer's Responses to Questions

**Comments to the Author**

1. Is the manuscript technically sound, and do the data support the conclusions?

Reviewer #1: Partly

Reviewer #2: Partly

Reviewer #3: Yes

2. Has the statistical analysis been performed appropriately and rigorously? 

Reviewer #1: Yes

Reviewer #2: Yes

Reviewer #3: Yes

3. Have the authors made all data underlying the findings in their manuscript fully available?

Reviewer #1: Yes

Reviewer #2: Yes

Reviewer #3: Yes

4. Is the manuscript presented in an intelligible fashion and written in standard English?

Reviewer #1: Yes

Reviewer #2: Yes

Reviewer #3: Yes

5. Review Comments to the Author

Reviewer #1: Dear authors,

Congratulations for your effort in analyzing the current data related to this surgical procedure. There are some important issues that should be addressed before this paper is published:

1. Demographic data of the participants should be discussed. In the majority, this studies were conducted in asian populations. It is largely known that can have different patterns of anatomic characteristics in this population when compared to caucasian or hispanic origin population

2. I have doubts about the baseline and postoperative mean IOP used to perform the metanalysis (Figure 3), and because of that, maybe conclusion can be different

3. Authors delineated conclusions before discuss the heterogeneity of the compiled data

Reviewer #2: 1. Page 2, line 17: TSIA must be corrected to TISA.

2. Page 5, line 3: Which device(s) was/were applied for the secondary outcome, eg, UBM, AS-OCT?

3. P7, line 20: TSIA must be corrected to TISA.

4. Discussion: Limitations of this meta-analysis should be discussed.

5. P11: References 18 and 20 are not related to this meta-analysis study. Why did you cited them?

Reviewer #3: First of all, thank you for your contribution in field of glaucoma.

This manuscript entitled 'Phacoemulsification combined with goniosynechialysis versus phacoemulsification

alone for patients with primary angle-closure disease: a meta-analysis" were well written. This manuscript well summarized the pre-existing studies and results.

In general, this article well adhere to PRISMA 2020. In specific the title identify the report as meta-analysis and methods included the Cochrane review (RevMan 5.4). Introduction well describe the rationale for the review. Methods specified the inclusion and exclusion criteria well, and selection process looks like clear and Interpretation is not clinically biased.

(In aspect of Search strategy, Selection process, data collection process, Study risk of Bias assessment)

Although some included study has small sample size, the interpretation looks like technically sound. And the assessment of bias is adequate.

If possible, the explain the detail of benefits and possible risk of Phaco+GSL for PAC for general readers.

Thank you again for your contribution for meta-analysis for Phaco + GSL.

6. PLOS authors have the option to publish the peer review history of their article (what does this mean?). If published, this will include your full peer review and any attached files.

Reviewer #1: **Yes: **Christiane Rolim-de-Moura

Reviewer #2: No

Reviewer #3: No

---

## [Author Response · Author response to Decision Letter 0]

29 Nov 2023

Dear reviewer,

Thank you for your comments. The following is our replies.

For reviewer #1: 

Congratulations for your effort in analyzing the current data related to this surgical procedure. There are some important issues that should be addressed before this paper is published:

1. Demographic data of the participants should be discussed. In the majority, this studies were conducted in asian populations. It is largely known that can have different patterns of anatomic characteristics in this population when compared to caucasian or hispanic origin population.

Answer: We added limitations of this study in the discussion section.

2. I have doubts about the baseline and postoperative mean IOP used to perform the metanalysis (Figure 3), and because of that, maybe conclusion can be different.

Answer: In Figure 3, mean IOP was the change of IOP before and after surgery.

3. Authors delineated conclusions before discuss the heterogeneity of the compiled data.

Answer: Generally, if I2 was higher than 50%, indicating a high degree of heterogeneity, a random-effect model should be applied. If I2 was lower than 50%, indicating a low-to-moderate heterogeneity, both random-effect model and fixed-effect model could be applied. In our meta-analysis, we used a random-effect model for both low-to-moderate and high degree of heterogeneity. Therefore, we did not discuss the heterogeneity in the manuscript.

For Reviewer #2: 

1. Page 2, line 17: TSIA must be corrected to TISA.

Answer: We revised it.

2. Page 5, line 3: Which device(s) was/were applied for the secondary outcome, eg, UBM, AS-OCT?

Answer: Only AS-OCT was applied for AOD and TISA in the included studies. We added it in the method section.

3. P7, line 20: TSIA must be corrected to TISA.

Answer: We revised it.

4. Discussion: Limitations of this meta-analysis should be discussed.

Answer: We added limitations of this study in the discussion section.

5. P11: References 18 and 20 are not related to this meta-analysis study. Why did you cited them?

Answer: We have referred to the research methods of these articles, so we have cited them.

For Reviewer #3: 

First of all, thank you for your contribution in field of glaucoma.

This manuscript entitled 'Phacoemulsification combined with goniosynechialysis versus phacoemulsification alone for patients with primary angle-closure disease: a meta-analysis" were well written. This manuscript well summarized the pre-existing studies and results.

In general, this article well adhere to PRISMA 2020. In specific the title identify the report as meta-analysis and methods included the Cochrane review (RevMan 5.4). Introduction well describe the rationale for the review. Methods specified the inclusion and exclusion criteria well, and selection process looks like clear and Interpretation is not clinically biased.

(In aspect of Search strategy, Selection process, data collection process, Study risk of Bias assessment)

Although some included study has small sample size, the interpretation looks like technically sound. And the assessment of bias is adequate.

If possible, the explain the detail of benefits and possible risk of Phaco+GSL for PAC for general readers.

Thank you again for your contribution for meta-analysis for Phaco + GSL.

Answer: GSL is a safe surgical procedure. Compared with Phaco alone, the possible risk of Phaco+GSL is hyphema, which was reported in three included studies [9, 12, 13]. We have added the above content to the discussion section.

Best regards.

Sincerely

Haiqing Bai

---

## [Decision Letter · Decision Letter 1]

21 Dec 2023

Phacoemulsification combined with goniosynechialysis versus phacoemulsification alone for patients with primary angle-closure disease: a meta-analysis

PONE-D-23-24472R1

Dear Dr. Bai,

We’re pleased to inform you that your manuscript has been judged scientifically suitable for publication and will be formally accepted for publication once it meets all outstanding technical requirements.

Kind regards,

Andrzej Grzybowski

Academic Editor

PLOS ONE

Reviewers' comments:

Reviewer's Responses to Questions

**Comments to the Author**

1. If the authors have adequately addressed your comments raised in a previous round of review and you feel that this manuscript is now acceptable for publication, you may indicate that here to bypass the “Comments to the Author” section, enter your conflict of interest statement in the “Confidential to Editor” section, and submit your "Accept" recommendation.

Reviewer #2: All comments have been addressed

Reviewer #3: All comments have been addressed

2. Is the manuscript technically sound, and do the data support the conclusions?

Reviewer #2: Yes

Reviewer #3: Yes

3. Has the statistical analysis been performed appropriately and rigorously? 

Reviewer #2: Yes

Reviewer #3: Yes

4. Have the authors made all data underlying the findings in their manuscript fully available?

Reviewer #2: Yes

Reviewer #3: Yes

5. Is the manuscript presented in an intelligible fashion and written in standard English?

Reviewer #2: Yes

Reviewer #3: Yes

6. Review Comments to the Author

Reviewer #2: Page 17, please correct Aisa to Asia, otherwise the manuscript is accepted for publishing. Congratulation.

Reviewer #3: Dear Authors.

Thank you for insightful revision.

Your manuscript titled "Phacoemulsification combined with goniosynechialysis versus phacoemulsification alone for patients with primary angle-closure disease: a meta-analysis" might provide the insightful and clinically meaningful information for ophthalmologists.

7. PLOS authors have the option to publish the peer review history of their article (what does this mean?). If published, this will include your full peer review and any attached files.

Reviewer #2: No

Reviewer #3: No

---

## [Editor Report · Acceptance letter]

25 Jan 2024

PONE-D-23-24472R1 

PLOS ONE

Dear Dr. Bai, 

I'm pleased to inform you that your manuscript has been deemed suitable for publication in PLOS ONE. Congratulations! Your manuscript is now being handed over to our production team.

Kind regards, 

on behalf of

Dr. Andrzej Grzybowski 

Academic Editor

PLOS ONE